

# Seasonality of hydrological model spin-up time: a case study using the Xinanjiang model

Mohammad M. Rahman[1], Minjiao Lu[2,3] , Khin H. Kyi[2]

[1]Department of Environmental Science and Technology, Jessore University of Science and Technology, Jessore-7408, Bangladesh
[2]Department of Civil and Environmental Engineering, Nagaoka University of Technology, 1603-1, Kamitomioka, Nagaoka, Niigata 940-2188, Japan
[3]Dr. Eng., Adjunct Professor, Chongqing Jiaotong University, Chongqing, China

*Correspondence to:* Mohammad M. Rahman (romelku22@yahoo.com)

**Abstract.** The internal adjustment process of a hydrological model followed by an unusual initial condition is known as the model spin-up. And the time required for a complete adjustment is termed as the model spin-up time. Model results for the duration of this spin-up progression are greatly impacted by the initial conditions, and often impractical or erroneous. The speed of this adjustment process is affected by the characteristics of the input data sets and their persistence. This study discusses the variability and seasonality of hydrological model spin-up time against the aridity of the river basin using multi-year climatologies for 18 river basins distributed relatively snow-free regions of the USA. The Xinanjiang model was run with each of all available year input data sets with two extreme initial conditions (saturated and unsaturated) and thereafter detected the model equilibrium state based on the Mahalanobis distance between the soil moisture states of two model runs. The seasonality of model spin-up was investigated by conducting multiple simulations that start from different time of a year. The basin average soil moisture memory (SMM) timescale (Rahman et al., 2015) and basin aridity index was estimated and thereafter investigated their relationship with the average model spin-up time.

Analysis suggests that the spin-up time highly varies with the simulation starting time and the dryness of the river basin. Overall, in all basins, model achieves the equilibrium state quickly while the simulation starts in late autumn (October-November). On the other hand, model equilibrates slowly while simulation starts in spring (March-May). Wet basin shows stronger variability of the model spin-up time (mean range 154 days) throughout the year as compared with that of dry basins (mean range 78 days). The mean spin-up time is shorter for wet basins (154 days) and longer for dry basins (233 days). The spin-up times are 3-7 times longer than the SMM timescale. The basin-wise mean spin-up time shows linear and exponential relationship with the SMM timescale and the basin aridity index respectively. The relationship offers predictability of model spin-up time from widely available potential evaporation and precipitation data sets.

## 1 Introduction

When a model is calibrated with an unusual initial condition, the model undergoes some adjustment process to reach the normal equilibrium state (Yang et al., 1995; Cosgrove et al., 2003; de Goncalves et al., 2006; Rahman and Lu, 2015). The time required to complete this model adjustments or reaching its equilibrium condition in its internal states (i.e. soil moisture) is called as the model spin-up time. The length and behaviour of this spin-up process is a function of chosen initial conditions, model parameters and the model input variables (Seck et al., 2015). The model findings for the duration of this spin-up time is vastly affected by the initial condition, and often impractical or erroneous. The model outputs after its initial adjustments normally shows betteragreement with the observations and responds reasonably to the model inputs (Yang et al., 1995; Cosgrove et al., 2003; Seck et al., 2015). Consequently, it is important to pay particular attention to the model spin-up





process, its length and behaviour for the modellers. However, clear information about the length of spin-up time is often missing or model specific and cannot be applicable to all models (Rahman and Lu, 2015).

In practice, modellers tend to reduce this spin-up period or exclude the initial model outputs for improved modelling accuracies mostly guided by a guess. These techniques of reducing spin-up errors hold certain limitations. Therefore,

understanding the factors affecting the spin-up process and its behaviour is highly important for modelling communities. The effect of initial conditions in hydrological models have been investigated by several researchers (Goodrich et al., 1994; Senarath et al., 2000; Castillo et al., 2003; Zehe et al., 2005; Berthet et al., 2009; Nikolopoulos et al., 2011; Zhang et al., 2011; Minet et al., 2011). However, these studies were done highlighting either an event-scale or short-term response. Moreover, these literatures do not necessarily estimate the spin-up time or describe the principle to specify the equilibrium

condition of model state once it finishes the spin-up process (Seck et al., 2015).

Recently, few studies have discussed about the spin-up time and behaviour of integrated hydrological model (Ajami et al., 2014; Seck et al., 2015; Rahman and Lu, 2015). Rahman and Lu (2015) suggested an straightforward technique to estimate the maximum spin-up period of the Xinanjiang model (Zhao, 1992) under any state of initial conditions using only basin aridity index (ratio of annual potential evaporation to precipitation) information. Estimating maximum model spin-up time

would reduce uncertainity involves with initial conditions. In contrary, spin-up time of land surface models (LSMs) is well documented (Yang et al., 1995; Robock et al., 1998; Schlosser et al., 2000; Cosgrove et al., 2003; Rodell et al., 2005; Lim et al., 2012). These literatures mainly investigate the model spin-up behaviour under various settings of climate, vegetation and soil types. Reported spin-up time of LSMs varies from models to models and range from one to several years (de Goncalves et al., 2006; Yang et al., 1995; Chen and Mitchell, 1999; Cosgrove et al., 2003; Rodell et al., 2005). Despite that the

conclusions of these literatures are often model-specific, they provide important insights and guidelines for all modelling communities about the spin-up behavior.

Up-to-date spin-up studies are mostly done on the basis of a recursive model runs throughout a specific period (typically a single year) where the outputs at the end of one run turn out to be the initial conditions for the subsequent run (Yang et al., 1995). These single year recursive model runs are claimed to be eliminating year to year climatic variability and corresponds

any model adjustments from year to year exclusively to the spin-up process (Cosgrove et al., 2003). Recurrent annual forcing feds the identical temporal dynamics to the system and facilitates to distinguish between the effects of persistence in initial conditions (Seck et al., 2015). These recursive model runs actually tend to represent the actual climatology with only a single year forcing data (Cosgrove et al., 2003). Rahman and Lu (2015) tried to improve the representativity of this single year model run by analysing the model spin-up behaviour based on simulation results using three different climatological input

data sets (mean, 5th and 95th percentile of annual rainfall). However, these recursive model runs using a single year climatology may possibly be inadequate to prepare the model to act logically against all climatological phenomenon, and could be missing important additional insights. Moreover, this single year recursive model run always starts the simulation from a particular point of a year. Since, the spin-up process is said to be highly linked with the atmospheric forcing and surface conditions (Yang et al., 1995; Chen and Mitchell, 1999; Cosgrove et al., 2003; Rodell et al., 2005; de Goncalves et

al., 2006; Rahman and Lu, 2015), the spin-up behaviour would be different when the model simulations start from different times of a year. Keeping the same initial conditions and employing different starting climatology certainly affects spin-up process. Recently, Rahman et al. (2015) discussed about the seasonality of soil moisture memory (SMM). Therefore, it is intuitive for any model spin-up time to show certain seasonality.

This study attempted to analyse the seasonality of hydrological model spin-up time using the Xinanjiang model (XAJ)

(Zhao, 1992). The XAJ model is a conceptual hydrological model discussed in section 2.2. Unlike existing literature, this study uses multi-year climatology instead of a single year recursive model runs. This study believes that the use of multi-





year climatologies trains the model in a better way and the outcomes are more realistic. Moreover, to detect the seasonality of model spin-up time, it performs series of simulations that start from different times of a year (details are given in section 2.5). Using multi-year forcing climatology also requires the model equilibrium condition to be defined differently from the available spin-up literatures (Yang et al., 1995; Cosgrove et al., 2003; Rodell et al., 2005; Lim et al., 2012; Ajami et al.,

2014; Seck et al., 2015; Rahman and Lu, 2015). In recursive simulation based spin-up studies, the model equilibrium condition has been defined mainly based on the percent cutoff-based time (PC time). PC time is the time required for the yearly changes of model output (i.e. soil moisture state) to decline to a pre-defined threshold values (Cosgrove et al., 2003; de Goncalves et al., 2006). Since a single year climatology is repeated in a recursive simulation, the input value for a particular point of a year remains the same throughout the simulation. Therefore, the equilibrium conditions remains the

same for a particular point. The output value of December 31st becomes the initial conditions for January 1st for the next run. While simulation continues, the model adjusts the gap between the initial and the equilibrium conditions for a particular point of a year. This actually allows detecting the progress of adjustment process based on user defined resolution (i.e. daily, monthly, yearly). In contrary, using multi-year forcing employs varying temporal dynamics and the equilibrium state could be different from year to year. Therefore, it performs two simulations; one initialised with completely "unsaturated" (0% soil

moisture), and another initialised with completely "saturated" (100% soil moisture).

In XAJ model, the soil moisture is represented in three layers. When the XAJ model is run with two extreme initial conditions ("saturated" and "unsaturated"), the soil moisture stores of each simulation will gradually converge towards a common state of equilibrium, and thus would show negative correlations until it reaches the equilibrium state. This equilibrium model state can be detected by estimating Mahalanobis Distance (MD) (Mahalanobis, 1930) between the soil

moisture states (prognostic variable for this study) of two simulations. MD has been applied in many fields to solve the classification problems, where there are several groups and concerns of affinities between the groups are present (McLachlan, 1999; De Maesschalck et al., 2000). MD has been used to detect the outliers (Martens and Naes, 1992; Leroy and Rousseeuw, 1987), to select the calibration samples from a large set of measurements (Shenk and Westerhaus, 1991), to investigate the representativity between two data sets (Jouan-Rimbaud et al., 1997; Jouan-Rimbaud et al., 1998; Wilson and

Atkinson, 2007) and similarity between two river flow series (Corduas, 2011). MD is useful to measure the divergence or distance between groups in terms of multiple characteristics. MD weights the variables with their covariance, which attributes less weight to strongly correlated variables.

Running XAJ model using multi-year forcing climatologies declaring "saturated" and "unsaturated" initial conditions, this study investigated the seasonality of model spin-up time for 18 river basins across the USA. This study holds at least three

major comparative advantages over the existing spin-up literatures. Firstly, use of multi-year forcing allows to introduce inter-annual variability and overcome the limitations contains in single year recursive simulation in the sense of representativeness to the actual phenomenon. Secondly, it detects the model equilibrium state based on MD that is widely acceptable in the presence of co-linearity of datasets. Thirdly, it provides useful insights about the seasonality of model spin-up time that is missing in the available spin-up studies.

## 2 Materials and Methods

### 2.1 Study area

This study analyses 18 river basins across the USA. Stream gauge locations of the analysed river basins are shown in Fig. 1.

[Figure 1]





For the sake of consistency, this study opted to select the same river basins studied by Rahman et al. (2015) and Rahman and Lu (2015). Analysing same river basins allows the comparison between the model spin-up outcomes derived from two different methodologies. Moreover, it enables to relate the model spin-up time and soil moisture memory. Literature (Li and Lu, 2014; Lu and Li, 2014; Rahman, et al., 2005; Rahman and Lu, 2005) suggests that these river basins have good data records, and their discharge could be simulated by the XAJ model with good accuracies. The studied river basins are situated in almost snow-free regions. On the basis of 30-year climate normals (1981-2010) published by NOAA's National Climatic Data Center (freely accessible at http://www.ncdc.noaa.gov/oa/climate/normals/usnormals.html; accessed on November 13, 2013), the studied river basins do not observe more than 7 snow-days (a snow-day is a day that receives at least 2.5mm snow/day) and record less than 200mm of total new snow annually. Physical and hydro-climatic characteristics of the studied river basins' is summarized in Table 1.

[Table 1]

### 2.2 Xinanjiang model

The Xinanjiang model (XAJ) is a conceptual hydrological model (Zhao, 1992). The XAJ model was developed by the Flood Forecast Research Laboratory of the East China Technical University of Water Resources (presently Hohai University). In the XAJ model the runoff is formedbased on the repletion of storage concept. The runoff starts to propagateafter the unsaturated zone reaches its field capacity(spatially distributed), and afterwards produces runoff equivalent tothe rainfall excess with no further loss (Zhao, 1992). Inputs to the XAJ model are areal mean precipitation and potential evaporation and provides discharge from the whole basin as the output. Throughout this article, the iput data sets designate time series of daily precipitation and potential evaporation. This XAJ model is widely used in humid and semi-arid areas of China(Lu and Li, 2014). The model is strong in physical meaning and its parameter (15 in total) can be estimated by basin characteristics. In addition to discharge data, the XAJ model simulates a time series of soil moisture data as the internal model state.

### 2.3 Data

The basin scale daily precipitation, *P* (calculated based on the ground based daily mean aerial precipitation), potential evaporation, *PE* (estimated from NOAA Evaporation Atlas) and discharge, *Q* data (estimated from USGS hydro-climatic data) acquired from the U.S. Model Parameter Estimation Project (MOPEX) data set (Schaake et al., 2006) were used in this study. The data set is freely available at ftp://hydrology.nws.noaa.gov/ (accessed on 19 October 2013).

### 2.4 XAJ model parameters, calibration and validation

The XAJ model was run with the support of a web-based application (accessible at http://lmj.nagaokaut.ac.jp/~khin/; last accessed on May 09, 2016) (Kyi, 2016). This web interface assists the user to calibrate and run the XAJ model in a user friendly environment. Additionally, it provides helpful suggestion in parameter settings for calibration based on Li and Lu (2014). Moreover,it enables to visualise the hydrograph and calculates Nash-Sutcliffe (NASH) efficiency (Nash and Sutcliffe, 1970). NASH efficiency was calculated based on Eq. (1). The XAJ model parameters and their calibrated values are presented in Table 2.

$$NASH = 1 - \frac{\sum_{t=1}^{n}(Q_o(t) - Q_s(t))^2}{\sum_{t=1}^{n}(Q_o(t) - \bar{Q}_o)^2} \tag{1}$$

Where, $Q_o$, $Q_s$ and $\bar{Q}_o$ are the observed discharge, simulated discharge and average observed discharge respectively.

[Table 2]



### 2.5 Simulation design

Unlike recursive simulation with a single year climatologies, this study run the XAJ model using full length available observed data sets with two initial conditions (saturated and unsaturated). To detect the seasonality of model spin-up time, this study performs a series of XAJ model runs with varying simulation starting time. The first simulation started from the

1st of January, 1st year and the successive simulations were done with a simulation loop that shifts the simulation starting time by 10-days forward until it completes the loop (in this study, the loop completes at 21st December of last year). To maintain the consistency in length of the input data sets among the simulations, the shifted climatologies are placed at the end of the input climatologies, thus total number of data records remains the same for every simulation. Figure 2 explains the input data loop that shifts 10-days climatology for a data records from 1st January 1948 to 31st December 1999. In every

step, the model was run twice using the same input file with two different initial conditions (saturated and unsaturated). Initially, the XAJ model was calibrated with saturated initial condition and thereafter the daily streamflow was validated against those of the observed by taking spin-up time long enough to avoid the effects of the initial condition. Finally, the calibrated parameter values were exercised for the subsequent simulations. NASH efficiency reported in Table 3 represents only the first simulation.

[Figure 2]

### 2.6 Definition of model spin-up time

This study assumes that the model achieves its equilibrium state when two sets of soil moisture state (from "saturated" and "unsaturated" simulation) become similar. The similarity is measured based on MD. The model is said to be in equilibrium state when the MD score is zero (0). The spin-up time is defined as the number of days required for the MD to approach zero

(0). The MD is calculated based on Eq. (2).

$$MD\left(\vec{x_s}, \vec{x_{us}}\right) = \sqrt{\left(\vec{x_s} - \vec{x_{us}}\right)^T S^{-1} \left(\vec{x_s} - \vec{x_{us}}\right)} \quad (2)$$

where $MD\left(\vec{x_s}, \vec{x_{us}}\right)$ is the Mahalanobis Distance between the random vectors $\vec{x_s}$ (states of three soil moisture layers from "saturated" simulation) and $\vec{x_{us}}$ (states of three soil moisture layers from "unsaturated" simulation). $T$ and $S$ is the matrix transpose operator and covariance matrix (non-singular) between $\vec{x_s}$ and $\vec{x_{us}}$ respectively.

### 2.7 Calculation of monthly and basin scale model spin-up time

A basin that has 52-year (1948-1999) long observed data requires approximately 1899 simulations which include two model runs each. The spin-up time was estimated for every simulation and grouped into months based on the simulation starting time. The monthly spin-up time was then computed by averaging all the spin-up times for the respective months. The basin average spin-up time is the arithmetic mean of all months (January to December).

### 2.8 Calculation of corresponding aridity index, basin aridity index and SMM timescale

The monthly spin-up time calculation was followed by the computation of aridity index for the corresponding spin-up period. The aridity index, ζ was calculated based on Li and Lu (2014). Aridity index of the corresponding spin-up period for all simulations were calculated, Eq. (3). Monthly means were calculated from the corresponding aridity index based on the



simulation starting time, Eq. (4). Finally, yearly corresponding aridity index was calculated by averaging monthly aridity index of a particular year, Eq. (5):

$$\zeta_c = \frac{PE_c}{P_c} \tag{3}$$

Where, $\zeta_c$, $PE_c$ and $P_c$ are the corresponding aridity index, corresponding potential evaporation and corresponding ground based areal precipitation during the model spin-up time respectively.

$$\zeta_{m,p} = \frac{\sum_{i=1}^{n} \zeta_{cp,i}}{n} \tag{4}$$

Where, $\zeta_{m,p}$, $\zeta_{cp,i}$ and $n$ are the monthly aridity index of month $p$, corresponding aridity index of $ith$ simulation starting at month $p$ and number of simulations that started at month $p$ respectively.

$$\zeta_{y,q} = \frac{\sum_{i=1}^{n} \zeta_{mq,i}}{n} \tag{5}$$

Where, $\zeta_{y,q}$, $\zeta_{mq,i}$ and $n$ are the yearly aridity index of year $q$, monthly aridity index of $ith$ month of year $q$ and number of months in a year respectively.

Since the basin scale $PE$ and $P$ data have been used as the input to the XAJ model, the same data were not utilized for
calculating the basin aridity index to avoid any inherent correlation with the model spin-up time. Aridity indices of the studied river basins'were approximated from an independent data sets. The aridity indices of 400 MOPEX river basins spread over the USA were interpolated to approximate the aridity index values ofthe studied river basins. This basin aridity index has been used to discuss the relationship with the model spin-up time. Consistent with Rahman et al. (2015) and Rahman and Lu (2015), the analysed river basins are grouped as "dry" and "wet" according to their aridity index values.
Basins that show aridity value of less than 0.9 are referred to as wet basins while, the reminder are called as dry basins in the following sections.

Basin-wise SMM timescale was computed after Rahman et al. (2015), Eq. (6).

$$\tau_{SMM} = 24.76\left(e^{1.25\,\zeta} - 1\right) \tag{6}$$

Where, $\tau_{SMM}$ and $\zeta$ are the soil moisture memory timescale in days and basin aridity index respectively.

## 3 Results and discussions

### 3.1 NASH efficiency and SMM timescale

The daily NASH efficiencies of the analysed basins suggest that the simulated streamflow has a good agreement with that of observed data sets. Basin-wise NASH efficiency and SMM timescales are reported in Table 3.

[Table 3]

### 3.2 XAJ model spin-up time and corresponding aridity index

The spin-up time ranged from 1 to 1265 days. The corresponding aridity index ranged from 0.002 to 2.16. On the basis of simulation starting time, spin-up times are plotted against the corresponding aridity index of that spin-up period in Fig. 3. Figure 3 suggests an an exponential relationship between spin-up time and the corresponding aridity index. Although the relationship looks weaker in summer months, all the relationships are statistically significant at 0.0001% (N>2600). The



regression equations representing the relationship between month-wise spin-up times and corresponding aridity index are presented in Table 4.

[Figure 3]

[Table 4]

Mean monthly spin-up times disclose a distinct variations in wet ($\zeta<0.9$) and dry ($\zeta>0.9$) basins (Fig. 4). In wet basins, the XAJ model required longer time to be equilibrated when the model simulation started from the spring months (March-May), while it achieved equilibrium quickly for late autumn or early winter (October-December). In contrast, in dry basins, the XAJ equilibrated quickly in early spring (March-April) and autumn (August-October) and it took longer time for the equilibrium in late spring to summer (May-June). Overall in all basins spin-up time is highest in Spring (March-May) and lowest in late autumn (October-November). This implies that starting simulation from the beginning of a hydrological year (1 October for the US river basins) could save the spin-up time.

[Figure 4]

Figure 5 represents the association between yearly model spin-up times and corresponding aridity index of three river basins with different aridity indices. Yearly spin-up time is strongly correlated with the corresponding aridity index. The relationship is consistent in all years and under different climatic conditions. This implies that model spin-up process is mainly influenced by the aridity index during the corresponding spin-up period.

[Figure 5]

Since the model equilibrium state is defined on the basis of the model internal soil moisture state, theoretically, it is believed that the model spin-up time is principally influenced by the persistence characteristics of soil moisture. A low SMM is indicative towards a shorter lived soil moisture anomalies and fades away quickly to reach in equilibrium state. The shorter the memory the shorter the spin-up period. Basin-wise SMM timescale and the model spin-up time shows strong correspondence with $R^2$=0.79 (Fig. 6).

[Figure 6]

Both spin-up time and SMM timescale represent the behaviour of soil moisture. Analysis indicates that model spin-up times are much longer than SMM timescales. In this study spin-up time was estimated as 3-7 times longer than SMM timescale. Yang et al. (1995) showed that the spin-up time of PILPS (Project for Intercomparison of Land Surface Parameterization Scheme) experiment is three times larger than the e-folding time (SMM as defined in this article) at 0.1% PC threshold. In another LSM study, Cosgrove et al. (2003) provided that the spin-up times are 2-17 times larger than SMM depending on the nature of initialisation and PC threshold. Although the methodologies and studied models are completely different, these literature agrees the overall comparative weight of SMM timescale and the spin-up time. The SMM timescale accounts the number of lag days that required for the soil moisture autocorrelations to drop below the threshold significance at 95% confidence level. Therefore, in SMM timescale calculation, a complete shedding of soil moisture anomalies is not counted. On the other hand, in this spin-up analysis a complete equilibrium was achieved, thus required longer times.

The model spin-up time is shorter and varies largely (mean range is 154 days) for wet basins throughout the year. On the other hand, in dry basins, the spin-up time is relatively long and varies moderately (mean range 78 days) from month to month. Basin-wise monthly mean spin-up times are presented in Table 3. The overall spin-up time is shorter in wet basins (wet basins mean 148 days) than those of dry basins (dry basins mean 233 days). This is consistent with Rahman et al. (2015); Rodell et al. (2005); Lim et al. (2012); Cosgrove et al. (2003).




Although the relationship of model spin-up time with corresponding aridity index has been revealed, this study also choose to investigate the connection with the basin aridity index. Because of the implicit nature of corresponding aridity index, it is difficult to use in hydrological modeling practice. On the other hand, basin scale annual $PE$ and $P$ data sets are widely available and basin aridity index could easily be estimated. The XAJ model spin-up time discloses an exponential association with basin aridity index (interpolated from separatedata sets) with a $R^2 = 0.93$ (Fig. 7). Equation 7 shows the mathematical relationship between the model spin-up time and the basin aridity index .

$$\tau_{SP} = 64.88 \left( e^{1.44\,\zeta} - 1 \right) \tag{7}$$

where $\tau_{SP}$ is the XAJ model spin-up time in days and $\zeta$ is the basin aridity index.

[Figure 7]

Rahman et al. (2015) and Rahman and Lu (2015) argued that model reaches its equilibrium state rapidlyunder relatively wetter or low SMM condition. This study also consider that the XAJ model will achieve its equilibrium state very quickly and take little or no timeunder extremely wet conditions (aridity index approaches zero). Based on this assumption, the regression equation, Eq. (7) was optimised for further knowledge of model's behaviour under climatic conditions outside the studied basins. This relationship could be handy for a rough guess of the XAJ model spin-up time and may possibly be useful for simulations with wider confidence.

## 4. Conclusions

When a model is calibrated with an unusual initial condition, the model goes through some spin-up process to achieve its normal equilibrium state. Model results for the duration of this spin-up progression are greatly impacted by the initial conditions, and often impractical or erroneous. Therefore, understanding this spin-up period has been the interest of modelling communities, particularly for the LSMs. Most spin-up studies are done based on a recursive simulations using a single year climatologies. Arguably, conclusions based on this recursive model runs might be erroneous due the lack of representativeness in the climatological extremes within the single year climatology. Moreover, researchers used different thresholds to define the model equilibrium conditions, and thus lost the comparability or uniformity. Furthermore, recursive simulations based spin-up outcomes does not provide any insight about the seasonality of spin-up time.

Aiming to solve these limitations, this study detects and analyses the seasonality of spin-up time using multi-year climatologies adopting new techniques of model equilibrium definition. The spin-up time shows high seasonality and mainly controlled by the aridity index of model forcing. This analysis suggests that model spin-up time could vary based on the simulation starting time of a year. The simulation that starts from month of January might achieves the equilibrium quickly as compared that starts from the month of May. However, this conclusions are based on the American climatic conditions and it might show different seasonal cycles elsewhere.

The XAJ model spin-up time discloses an exponential association with the basin aridity index. This relationship permits approximating the XAJ model spin-up time utilising precipitation and potential evaporation data only. However, it is important to be noticed that this equation was derived on the basis of a daily scale model simulation, therefore the XAJ model spin-up time for different time scale could not be the same. Approximating the XAJ model spin-up time would eliminate the uncertainty linked with guessing, simply banking on feeling or experience. An advance hint of model spin-up time could enable us fullyutilization of the information contained in a shorter data records under inadequate data availability.



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





Table1: Studied MOPEX basins, locations and basic characteristics. Dry basins (ζ> 0.9) are marked with bold face, Italic font style.

| MOPEX ID | Location | | | Ave. P (mm/year) | Ave. PE (mm/year) | Ave. snow-days (day/year) | Ave. total new snow (mm/year) | Ave. soil moisture saturation (%) |
|---|---|---|---|---|---|---|---|---|
| | Long. | Lat. | State | | | | | |
| 11532500 | -124.05 | 41.79 | CA | 2687 | 740 | 0.00 | 0 | 82 |
| 12027500 | -123.03 | 46.78 | WA | 1599 | 579 | 3.00 | 127 | 75 |
| 03504000 | -83.62 | 35.13 | NC | 1893 | 762 | 3.90 | 193 | 90 |
| 03410500 | -84.53 | 36.63 | TN | 1389 | 817 | 6.20 | 160 | 74 |
| 02387500 | -84.94 | 34.58 | GA | 1480 | 901 | 0.70 | 18 | 73 |
| 03574500 | -86.31 | 34.62 | AL | 1467 | 941 | 0.80 | 41 | 74 |
| 07378500 | -90.99 | 30.46 | LA-MS | 1594 | 1077 | 0.60 | 23 | 63 |
| 07375500 | -90.36 | 30.51 | LA-MS | 1633 | 1074 | 0.60 | 23 | 64 |
| 02492000 | -89.90 | 30.63 | LA-MS | 1583 | 1071 | 0.60 | 23 | 47 |
| 02456500 | -86.98 | 33.71 | AL | 1425 | 982 | 0.80 | 41 | 66 |
| 02472000 | -89.41 | 31.71 | MS | 1492 | 1060 | 0.60 | 23 | 64 |
| 07290000 | -90.70 | 32.35 | MS | 1435 | 1073 | 0.60 | 23 | 57 |
| 07056000 | -92.75 | 35.98 | AR | 1180 | 916 | 3.80 | 132 | 68 |
| 07288500 | -90.54 | 33.55 | MS | 1381 | 1112 | 0.60 | 23 | 62 |
| *07072000* | *-91.11* | *36.35* | *AR* | *1114* | *964* | *3.80* | *132* | *62* |
| *07197000* | *-94.84* | *35.92* | *OK* | *1162* | *1113* | *5.6* | *198* | *58* |
| *08033500* | *94.40* | *31.02* | *TX* | *1100* | *1308* | *1.30* | *4* | *60* |
| *06914000* | *-95.25* | *38.33* | *KS* | *957* | *1206* | *10.00* | *373* | *61* |



Table 2. Calibrated parameter values in the Xinanjiang model.

| Parameter | Physical meaning | Range |
|---|---|---|
| $C_p$ | Ratio of measured precipitation to actual precipitation | 0.92-1.1 |
| $C_{ep}$ | Ratio of potential evapotranspiration to pan evaporation | 0.9-1.29 |
| $b$ | Exponent of the tension water capacity curve | 0.1-0.3 |
| $imp$ | Ratio of the impervious to the total area of the basin | 0-0.0001 |
| $WUM$ | Water capacity in the upper soil layer (mm) | 20 |
| $WLM$ | Water capacity in the lower soil layer (mm) | 50-90 |
| $WDM$ | Water capacity in the deeper soil layer (mm) | 20-80 |
| $C$ | Coefficient of deep evapotranspiration | 0.1-0.3 |
| $SM$ | Areal mean free water capacity of the surface soil layer (mm) | 5-55 |
| $EX$ | Exponent of the free water capacity curve | 0.5-1.5 |
| $KI$ | Outflow coefficient of the free water storage to interflow | 0.1-0.65; KI+KG=0.7 |
| $KG$ | Outflow coefficient of the free water storage to groundwater | 0.08-0.6; KI+KG=0.7 |
| $c_s$ | Recession constant for channel routing | 0.5-0.88 |
| $c_i$ | Recession constant for the lower interflow storage | 0.3-0.82 |
| $c_g$ | Daily recession constant of groundwater storage | 0.982-0.998 |





Table 3. Summary of the XAJ model spin-up time analysis. Dry basins ($\zeta > 0.9$) are marked with bold face, Italic font style.

| MOPEX ID | Area (sq.km) | Data length (year) | Daily NASH | Basin aridity index ($\zeta$) | $\tau_{SMM}$ (day) | $\tau_{SP}$ (day) | | |
|---|---|---|---|---|---|---|---|---|
| | | | | | | Min. (month) | Max. (month) | Mean |
| 11532500 | 1577 | 52 | 0.79 | 0.29 | 11 | 18 (Nov) | 156 (May) | 72 |
| 12027500 | 2318 | 52 | 0.81 | 0.39 | 16 | 25 (Nov) | 205 (Apr) | 98 |
| 03504000 | 135 | 52 | 0.81 | 0.40 | 16 | 36 (Dec) | 147 (Apr) | 83 |
| 03410500 | 2471 | 52 | 0.65 | 0.58 | 26 | 51 (Nov) | 227 (Mar) | 137 |
| 02387500 | 4144 | 52 | 0.78 | 0.61 | 28 | 58 (Dec) | 224 (Dec) | 135 |
| 03574500 | 829 | 52 | 0.65 | 0.64 | 30 | 48 (Nov) | 216 (Apr) | 121 |
| 07378500 | 3315 | 51 | 0.66 | 0.70 | 35 | 100 (Nov) | 154 (Apr) | 185 |
| 07375500 | 1673 | 51 | 0.67 | 0.71 | 35 | 134 (Oct) | 295 (Mar) | 216 |
| 02492000 | 3142 | 52 | 0.61 | 0.71 | 36 | 69 (Nov) | 222 (May) | 155 |
| 02456500 | 2292 | 52 | 0.80 | 0.72 | 36 | 74 (Dec) | 246 (Apr) | 153 |
| 02472000 | 1924 | 52 | 0.71 | 0.76 | 39 | 69 (Nov) | 215 (Apr) | 144 |
| 07290000 | 7283 | 50 | 0.67 | 0.80 | 43 | 116 (Nov) | 258 (Mar) | 197 |
| 07056000 | 2147 | 52 | 0.66 | 0.81 | 43 | 82 (Oct) | 230 (Mar) | 169 |
| 07288500 | 1987 | 42 | 0.70 | 0.86 | 48 | 135 (Oct) | 272 (Mar) | 206 |
| *07072000* | *1134* | *46* | *0.71* | *0.90* | *52* | *184 (Oct)* | *257 (Jun)* | *216* |
| *07197000* | *795* | *52* | *0.65* | *0.94* | *55* | *145 (Sep)* | *232 (Jun)* | *193* |
| *08033500* | *9417* | *52* | *0.61* | *1.19* | *85* | *205 (Nov)* | *305 (May)* | *250* |
| *06914000* | *865* | *52* | *0.62* | *1.34* | *108* | *237 (Aug)* | *290 (Jan)* | *273* |





Table 4. Relationships between month-wise model spin-up time and corresponding aridity index.

| Month | Equation ($\tau_{sp}$ = model spin-up time, $\zeta$ = aridity index) | Coefficient of determination ($R^2$) | Number of data points (N) |
|---|---|---|---|
| January | $\tau_{sp} = 26.22\, e^{2.17\, \zeta}$ | 0.77 | 3052 |
| February | $\tau_{sp} = 22.34\, e^{2.16\, \zeta}$ | 0.70 | 2749 |
| March | $\tau_{sp} = 30.97\, e^{1.85\, \zeta}$ | 0.53 | 2953 |
| April | $\tau_{sp} = 55.97\, e^{1.24\, \zeta}$ | 0.30 | 2706 |
| May | $\tau_{sp} = 98.23\, e^{0.67\, \zeta}$ | 0.13 | 2658 |
| June | $\tau_{sp} = 86.24\, e^{0.76\, \zeta}$ | 0.16 | 2617 |
| July | $\tau_{sp} = 78.08\, e^{0.76\, \zeta}$ | 0.16 | 2806 |
| August | $\tau_{sp} = 41.50\, e^{1.48\, \zeta}$ | 0.37 | 2822 |
| September | $\tau_{sp} = 31.10\, e^{1.97\, \zeta}$ | 0.54 | 2874 |
| October | $\tau_{sp} = 30.37\, e^{2.17\, \zeta}$ | 0.67 | 3050 |
| November | $\tau_{sp} = 29.36\, e^{2.23\, \zeta}$ | 0.69 | 3005 |
| December | $\tau_{sp} = 29.91\, e^{2.12\, \zeta}$ | 0.75 | 3052 |



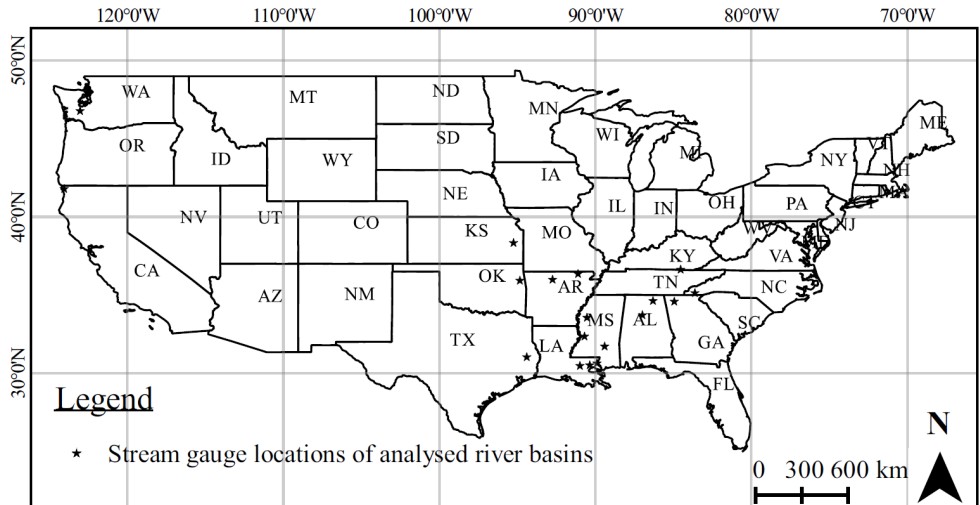

**Figure 1: Stream gauge location map of studied river basins over USA mainland.**



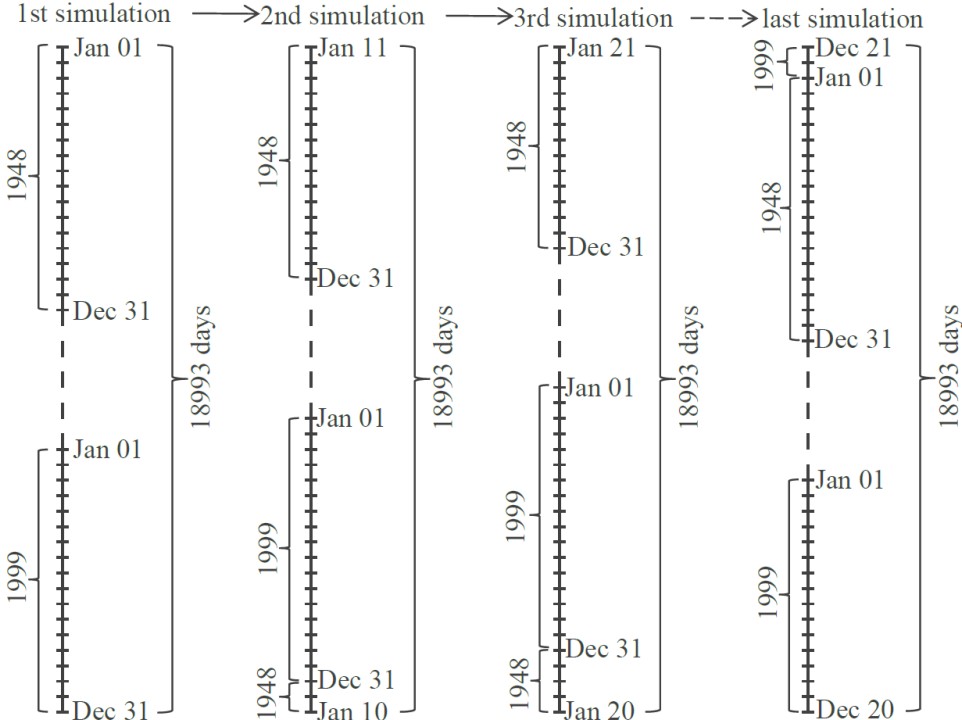

**Figure 2: Illustration of the input data loop for XAJ model simulation**.





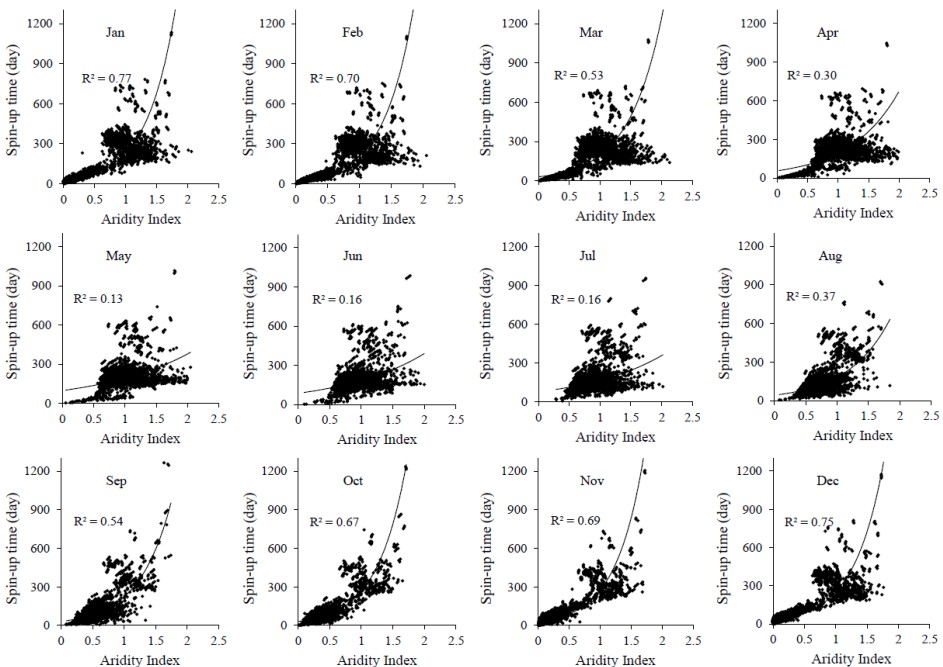

Figure 3: XAJ model spin-up times and their corresponding aridity index plotted based on the simulation starting time.





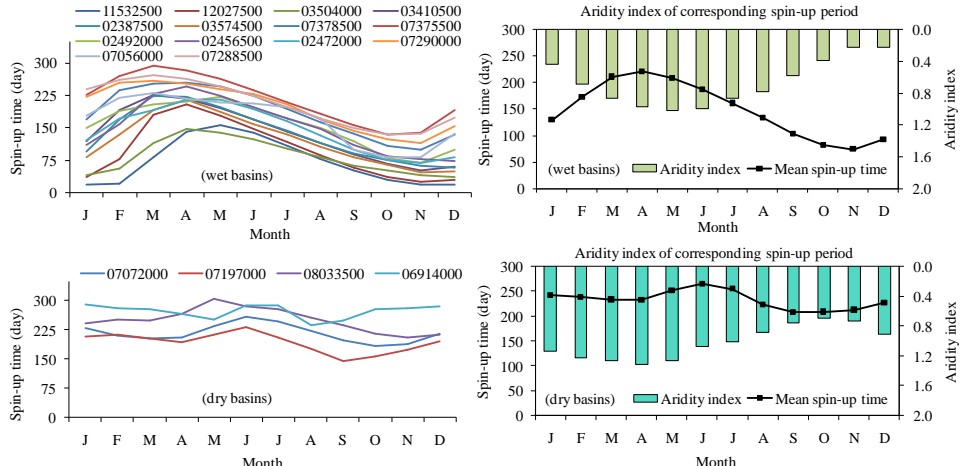

**Figure 4: Mean monthly model spin-up time and the corresponding aridity index for wet basins (ζ< 0.9) and dry basins (ζ>0.9).**





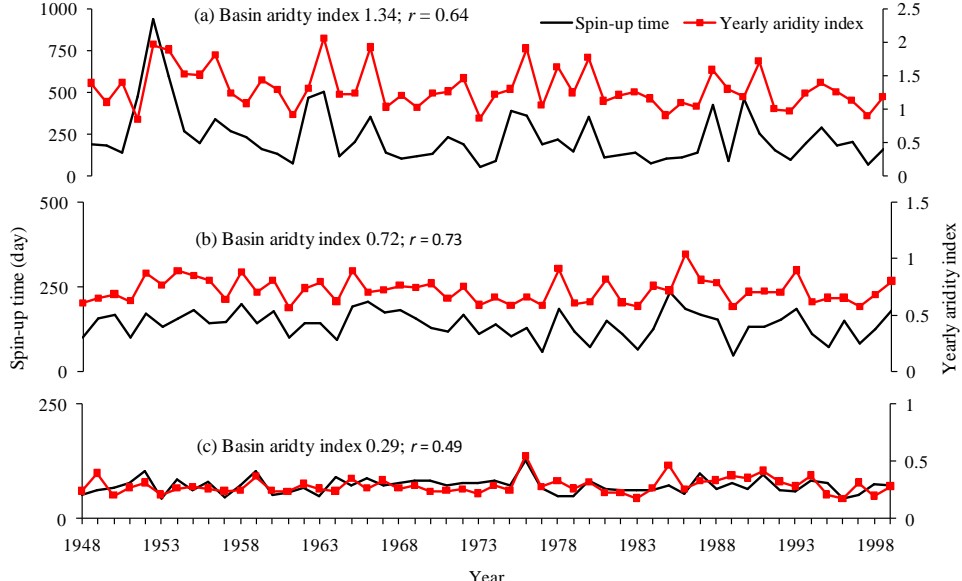

**Figure 5: Yearly model spin-up time and the aridity index. r is the correlation coefficient between yearly model spin-up time and aridity index.**



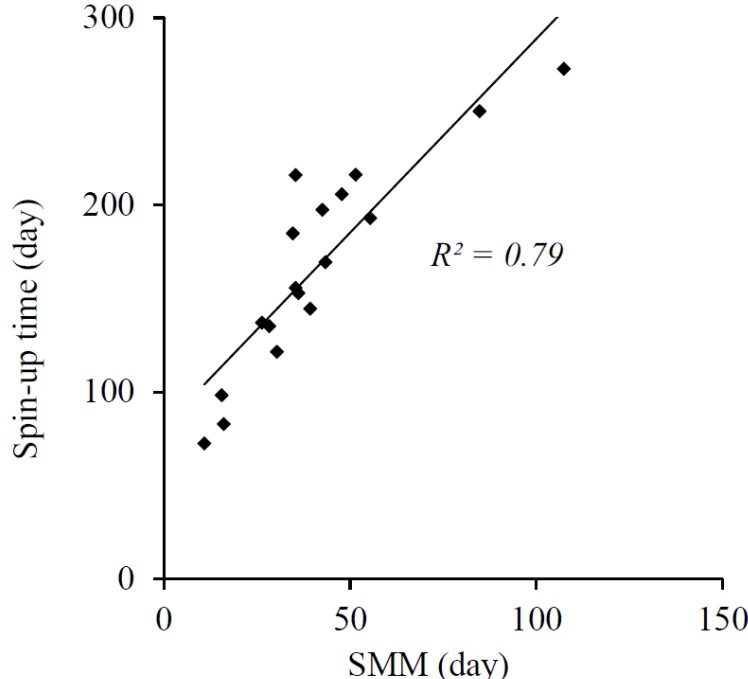

**Figure 6: SMM timescale and the model spin-up time.**




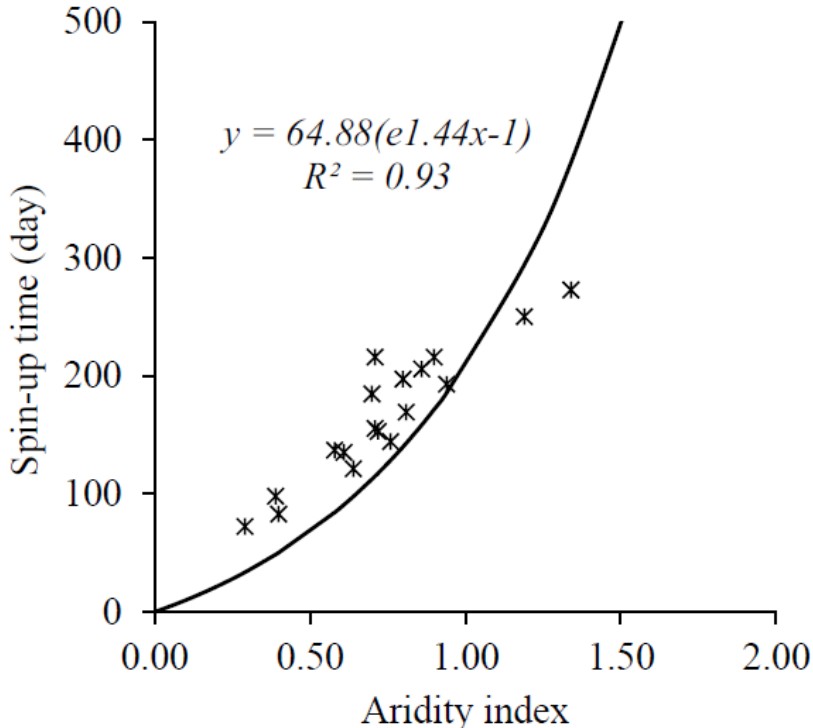

**Figure 7: Relationship between XAJ model spin-up time and basin aridity index.**

