# Peer review of "Seasonality of hydrological model spin-up time: a case study using the Xinanjiang model"

_Hydrology and Earth System Sciences, 2016_

## Referee Comment (RC1) · Anonymous Referee #1 · 16 Sep 2016

Authors estimated spin-up time of a conceptual hydrologic model by initializing the model from two extreme initial conditions (wet and dry states) at different times of a year, and defining the equilibrium state by computing the Mahalanobis distance between the soil moisture states of two model runs. By initializing the model at different times of a year, authors were able to present variability of spin-up time from models started from different seasons. While model initialization is an important issue in hydrologic modelling and erroneous initial conditions lead to bias on simulated model output(s), there are a few issues with the current study as highlighted below.

1) The choice of a modelling application by the authors does not seem to be relevant for a spin-up study. As it is shown in the results, spin-up time of a conceptual lumped

hydrologic model is quite short in the study catchments. In fact, the impact of initial condition is mostly disappeared in less than a year or two. In the conceptual hydrologic modelling literature, we often see that the first 2 or 3 years of simulation is considered as the warm-up period and removed from the rest of the analysis including model calibration. Therefore, it is not clear what are the implications of estimating spin-up time in conceptual hydrologic models considering their short spin-up times? Can we just remove the first two years of simulation as it is commonly performed? I agree with the authors that initialization is an important issue in hydrologic modelling. However, proper initialization of hydrologic models becomes more relevant in integrated hydrologic models where sub-surface and land surface processes are coupled and computational time is an important issue.

2) Another issue with the approach in the paper is that it seems model calibrations are performed at the same time to explore the impact of initial condition. Therefore, one would expect that the impact of initial condition is reduced through parameter adjustment during calibration. It will be very useful if authors can present possible differences in model parameters and performance between the various initializations as the model reaches equilibrium.

3) Why authors only used snow free catchments in their modelling study. The conceptual model used here runs quite fast and it is not clear why not all the MOPEX data is used in this study. At least this way, authors could expand their analysis and provide useful and informative discussion that will be useful for readers.

4) As authors have stated in their Introduction, Rahman and Lu (2015) have already estimated the maximum spin-up period of the Xinanjiang model using basin aridity index. Similarly seasonality of soil moisture memory (SMM) is already discussed in Rahman et al. (2015). Therefore, the novelty of this manuscript is not clear.

5) No information is provided about the seasonal characteristics of precipitation in the study basins. I suspect that seasonality of spin-up time is similar to the seasonality of

precipitation patterns. Can authors discuss this?

6) It will be very important that authors discuss their results and explain the factors impacting observed behaviour here. More importantly, authors should discuss the implications of their study and its broader impact for hydrologic modelling community.

7) Regarding the calibration, no information is provided about the calibration approach and the length of calibration and evaluation periods.

8) Why spin-up time is longer if the start time of a simulation is in spring? Can authors provide further details.

9) In Page 2 –Line 4: authors state "These techniques of reducing spin-up errors hold certain limitations". Can authors discuss these limitations? There are multiple papers that examined the use of various spin-up criteria in both land surface and integrated hydrologic models and it is not clear what authors are referring to it here.

––––––––––––––––––––––

---

## Referee Comment (RC2) · Anonymous Referee #2 · 21 Sep 2016

General Comments:

The authors present an analysis of model spinup time (defined here as the number of days for fully saturated and fully dry model initializations to converge) as a function of model run start season and modeled basin aridity. To explore this spinup time variability, they apply the conceptual Xinanjiang hydrologic model to 18 MOPEX basins, mostly located in the South-Central U.S. They find statistically significant relationships between aridity index and model spinup time, as well as variability in required spinup time depending on the start season of the model run. They also find that the seasonality of spinup time varies for "wet" and "dry" basins. Their study culminates in a well-defined exponential relationship between aridity index and spinup time that potentially has relevance for other applications using a similar model configuration.

The authors make a nice case for the importance of model spinup in hydrologic modeling studies and the lack of consensus in quantitative methods and strategies for assessing model spinup. Their research question on the variability in spinup time based on climatic setting is an interesting one. However, I don't think the study in its current state addresses this question in a way that is relevant to the broader hydrologic modeling community. Specifically, I believe the authors should address the following key points:

1) What is the authors' working hypothesis for this study? What is the conceptual basis for expecting spinup time to vary based on aridity and season? Why focus on these two factors and not others, such as geophysical or biological conditions?

2) Model spinup can be a significant burden for hydrologic models covering large, distributed regions or highly complex physical process-based models, both of which can be computationally intensive. Lumped and conceptual models are often less expensive to run and therefore spinup time is less of a concern. I recommend that the authors make a better case as to why it is appropriate to use the conceptual Xinanjiang model applied to a small subset of individual basins to answer the broader question of what controls spinup time. If we do not expect spinup times to be similar for conceptual and physical models, would we expect the same climatic or environmental controls?

3) Following from (1) and (2), there are systems that are known to require longer spinup times, such as deep groundwater aquifers, large surface storages, etc. Is the study model configuration capturing any of these slower processes, or purely focused on shallow soil water storage? How appropriate is this model configuration for the study basins' dominant hydrologic regime? How generally applicable are the findings if these processes are not represented? I recommend the authors provide a bit more detail on the model and its appropriateness for the study basins.

4) How does the model calibration affect the results? The calibration procedure is a bit

difficult to understand based on the description provided, so I recommend going into a bit more detail on what was done and how it may/may not be impacting spinup times.

5) The authors present some interesting patterns in seasonality of spinup time. However, there is little discussion of the potential physical reasons for the patterns. Are these patterns simply mirroring seasonal patterns of precipitation? Are there other physical or climatic controls that might explain some of the spread, or help us determine how we can apply these results to other models or domains? I recommend expanding the discussion section to address some of these questions, which should give the paper a much broader relevance.

Specific Comments:

Overall: The paper is nicely organized and figures are clear. However, it would benefit from additional grammar/typo editing throughout the paper.

1. Introduction: Portions of the introduction section (e.g., lines 16-27) read more like methods than introduction. I would recommend moving these specifics on the model strategy to the methods section and dedicate a bit more of the intro to addressing the study hypotheses and rationale.

2. Materials and Methods 2.1 Study area: Why choose only snow-free basins? I would guess because the model does not represent snowpack dynamics, but this should be stated. Why these particular 18 basins? The basins are primarily in the South-Central US (with 2 exceptions), not distributed across the US. I recommend describing the hydrologic regime in this region so we understand some of the seasonal patterns better – what is the seasonality of precipitation? Is there deep groundwater storage? What controls the runoff response?

2.2 Xinanjian Model: The model assumptions and configuration are fairly important for this study, so I think more detail on the model description is warranted. For example, it is not clear whether this is a lumped or distributed model when applied to the individual

basins. What types of hydrologic systems does the model perform well in, and what types do poorly with this conceptual representation?

2.4 Parameters, Calibration, Validation: The calibration/validation procedure is not really described here, and is only vaguely described in the next section. I recommend expanding this section to detail the calibration/validation procedure so the reader can understand the potential sensitivity of the spinup results to the calibration.

2.5 Simulation Design: Per the previous comment, it is difficult to disentangle the calibration procedure and the spinup procedure based on the description provided. I recommend separating the descriptions and clarifying the procedures.

3. Results and Discussion: Per the general comment, I recommend adding discussion on the possible physical reasons for some of the observed patterns. As written, this section is really just results. SMM and its calculation should be better defined. The results really need to be related back to the model assumptions, climate regime, or physical basin characteristics to be relevant to other studies.

---

## Author Comment (AC1) · 21 Oct 2016

**Seasonality of hydrological model spin-up time: a case study using the Xinanjiang model (doi:10.5194/hess-2016-316)**

**Author response for reviewer comments #1**

1) The choice of a modelling application by the authors does not seem to be relevant for a spin-up study. As it is shown in the results, spin-up time of a conceptual lumped hydrologic model is quite short in the study catchments. In fact, the impact of initial condition is mostly disappeared in less than a year or two. In the conceptual hydrologic modelling literature, we often see that the first 2 or 3 years of simulation is considered as the warm-up period and removed from the rest of the analysis including model calibration. Therefore, it is not clear what are the implications of estimating spin-up time in conceptual hydrologic models considering their short spin-up times? Can we just remove the first two years of simulation as it is commonly performed? I agree with the authors that initialization is an important issue in hydrologic modelling. However, proper initialization of hydrologic models becomes more relevant in integrated hydrologic models where sub-surface and land surface processes are coupled and computational time is an important issue.

**Author response:** The importance of estimating model spin-up time is discusses in Rahman and Lu (2015).

We agree that conceptual lumped hydrological model spin-up time is quite short, and thus excluding initial model outcomes (warm-up period) is an easy solution to avoid errors associated with model initialization. Of course, we can simply remove first two years of simulations while performing analysis.
However, in some cases exclusion of one year model out outputs could be a very costly task in developing countries where hydro-climatic data is very scarce (say only 2-3 years of available data records). Over-estimating the spin-up period would lead to a loss of important information. Likewise, an underestimation would affect the conclusion by incorporating erroneous initial model outputs. Moreover, guessing spin-up time (if any) for a shorter period, particularly for seasonal or monthly simulation would be very problematic (Rahman and Lu, 2015). Arguing these limitations, Rahman and Lu (2015) discusses the variations of model spin-up time with basin hydro-climatic characteristics and suggest an easy way to estimate maximum spin-up time under extreme hydro-climatic and data scarce conditions with improved accuracies.

The comparative advantages of this manuscript over Rahman and Lu (2015) have been discussed in the manuscript. Please see Page-3, Line 29-34.

2) Another issue with the approach in the paper is that it seems model calibrations are performed at the same time to explore the impact of initial condition. Therefore, one would expect that the impact of initial condition is reduced through parameter adjustment during calibration. It will be very useful if authors can present possible differences in model parameters and performance between the various initializations as the model reaches equilibrium.

**Author response:** The model calibration and exploration of model spin-up time were performed separately (please see page-5, line-11-14). The Xinanjiang model was calibrated with saturated initial condition and thereafter the daily streamflow was validated against those of the observed by taking spin-up time long enough (10 year) to avoid the effects of the initial condition. Thereafter, these calibrated parameter values were exercised for the subsequent simulations that explores the impact of initial condition.

3) Why authors only used snow free catchments in their modelling study. The conceptual model used here runs quite fast and it is not clear why not all the MOPEX data is used in this study. At least this way, authors could expand their analysis and provide useful and informative discussion that will be useful for readers.

**Author response:** The studied river basins were selected intentionally mainly for two reasons.

Firstly, to maintain consistency with Rahman and Lu (2015) and Rahman et al. (2015) as they discussed model-spin-up time and soil moisture memory for the same river basins. As we mentioned earlier that Rahman and Lu (2015) discussed model spin-up time based on a different methodologies (single year recursive simulation) that of ours (multiyear climatologies). We prefer to analyze same river basins for comparing model spin-up outcomes derived from two different methodologies. On the other hand, Rahman et al. (2015) analyzed soil moisture memory (SMM) for the same river basins. Therefore, it enables to relate the model spin-up time and soil moisture memory. Since, soil moisture autocorrelation equation (based on what SMM was estimated) does not consider snow, Rahman et al. (2015) choose snow free MOPEX basins for analysis and ultimately led the selection of studied basin for the present study.

Secondly, experience suggests that the XAJ model does not produce better results for all the MOPEX basins, particularly for the drier basins (Kyi, 2014).

4) As authors have stated in their Introduction, Rahman and Lu (2015) have already estimated the maximum spin-up period of the Xinanjiang model using basin aridity index. Similarly seasonality of soil moisture memory (SMM) is already discussed in Rahman et al. (2015). Therefore, the novelty of this manuscript is not clear.

**Author response:** This study holds at least three major comparative advantages over the existing spin-up literatures including Rahman and Lu, 2015. Firstly, use of multi-year forcing allows to introduce inter-annual variability and overcome the limitations contains in single year recursive simulation in the sense of representativeness to the actual phenomenon. Secondly, it detects the model equilibrium state based on Mahalanobis Distance that is widely acceptable in the presence of co-linearity of datasets. Thirdly, it provides useful insights about the seasonality of model spin-up time that is missing in the available spin-up studies. The novelty of this manuscript has been discussed elaborately in Page-2, Line 22-41 and Page-3, Line-1-15.

SMM and model spin-up time is similar but not necessarily the same. Rahman et al. (2015) shows the variability of SMM based on basin hydro-climatic characteristics. On the other hand, this manuscript discusses the variations of model spin-up time based on the simulation starting time (and of course due to variations of hydro-climatic conditions). Understanding seasonality of both the phenomenon certainly improves the understanding of the hydrological research communities.

5) No information is provided about the seasonal characteristics of precipitation in the study basins. I suspect that seasonality of spin-up time is similar to the seasonality of precipitation patterns. Can authors discuss this?

[Figure]

Fig. 1: Seasonality of precipitation and model spin-up time: (a) wet basins and (b) dry basins.

**Author response:** Thank you very much. You are right. Figure 1 suggests that, the seasonality of spin-up time is similar to the seasonality of precipitation. Monthly spin-up time shows strong correlation ($r=0.87$) with monthly total precipitation. The spin-up time was explored by declaring two extreme initial conditions (0% and 100% soil moisture). It is intuitive that the higher the distance between the mean condition and the initial condition, the longer the time requires reaching the equilibrium. The distance between the mean condition and any of the initial condition for the months with higher precipitation is wider as compared than that of months with lesser precipitation. Therefore, the higher the monthly precipitation, the larger the spin-up time is.

This discussion will be added into the manuscript.

6) It will be very important that authors discuss their results and explain the factors impacting observed behaviour here. More importantly, authors should discuss the implications of their study and its broader impact for hydrologic modelling community.

**Author response:** The study provides important information to the modeling community, particularly for those who works under data scarce situation. It reveals the seasonality of spin-up time and explains the control of spin-up behavior. Additionally, it suggests an easy way to estimate the spin-up time using only the precipitation and potential evapotranspiration information. Estimating spin-up time could improve modeling efficiency under data scarce situation.

7) Regarding the calibration, no information is provided about the calibration approach and the length of calibration and evaluation periods.

**Author response:** Please see author response of question no. 2.

8) Why spin-up time is longer if the start time of a simulation is in spring? Can authors provide further details.

**Author response:** In wet basins the spin-up time is longer while the simulation starts in spring. Figure 1 agrees that the spring months' precipitation is higher as compared that of other months. Please also see explanation in question no. 5.

9) In Page 2 –Line 4: authors state "These techniques of reducing spin-up errors hold certain limitations". Can authors discuss these limitations? There are multiple papers that examined the use of various spin-up criteria in both land surface and integrated hydrologic models and it is not clear what authors are referring to it here.

**Author response:** This is refers to the limitations contained in guessing spin-up time. The limitations are explained in author response to the question no. 1.

The explanation will be added into the manuscript.

References:

Kyi, K.H., Development of a user friendly web-based Xinanjiang model with calibration support system. Unpublished M.Sc. Thesis, Dept. of Civil and Environmental Engineering, Nagaoka University of Technology, Japan. 2014.

Rahman, M.M., Lu, M., and Kyi, K. H.: Variability of soil moisture memory for wet and dry basins, J. Hydrol., 523, 107–118, doi: 10.1016/j.jhydrol.2015.01.033, 2015.

Rahman, M.M. and Lu, M.: Model spin-up behavior for wet and dry basins: a case study using the Xinanjiang model, Water, 7, 4256–4273, doi: 10.3390/w7084256, 2015.

---

## Author Comment (AC2) · 21 Oct 2016

**Seasonality of hydrological model spin-up time: a case study using the Xinanjiang model (doi:10.5194/hess-2016-316)**

**Author response for reviewer comments #2**

1) What is the authors' working hypothesis for this study? What is the conceptual basis for expecting spinup time to vary based on aridity and season? Why focus on these two factors and not others, such as geophysical or biological conditions?

**Author response:** This study hypothesized that the model spin-up time does vary with timing of the simulation start.

Literature (i.e. Cosgrove et al., 2003; Rahman and Lu, 2015) suggests that soil moisture memory (SMM) time varies spatially and is correlated with precipitation and temperature. SMM affects the spin-up process. A low SMM indicates that the soil moisture anomalies are short-lived and dissipate hurriedly, enabling the model to recover relatively quickly from an undesirable initial state. On the other hand, a high SMM that indicates the slowness of anomaly dissipation and would delay the process of model equilibrium. Koster and Suarez, 2001 and Orth Et al., 2013 explained several controls of SMM (i.e. altitude, slope, soil cover, evapotranspiration, precipitation and runoff) and concluded precipitation and evaporation variability as the prominent control. Since SMM and spin-up time is analogous, this study also assumes that precipitation and evapotranspiration could also play dominant roles for dictating the spin-up time. The non-stationary nature of precipitation and evapotranspiration constructed the conceptual basis for expecting spinup time to vary based on aridity and season?

2) Model spinup can be a significant burden for hydrologic models covering large, distributed regions or highly complex physical process-based models, both of which can be computationally intensive. Lumped and conceptual models are often less expensive to run and therefore spinup time is less of a concern. I recommend that the authors make a better case as to why it is appropriate to use the conceptual Xinanjiang model applied to a small subset of individual basins to answer the broader question of what controls spinup time. If we do not expect spinup times to be similar for conceptual and physical models, would we expect the same climatic or environmental controls?

**Author response:** Considering the differences in model definition and structure, it is quite risky to consider the same control for all the models. Available spin-up literatures are mainly model specific. Lumped and conceptual models are less expensive to run for sure. However, spin-up time could be of a concern for data scarce situation or seasonal simulation (Rahman and Lu, 2015). Common practices consider first 2 or 3 years of simulation as the spin-up period and removes from the rest of the analysis including model calibration. The problem here is to define the spin-up period. This is usually done based on personal feeling, experience, available data records and purpose. In some cases exclusion of one year model outputs could be a very costly task in developing countries where hydro-climatic data is very scarce (say only 2-3 years of available data records). Over-estimating the spin-up period would lead to a loss of important information. Likewise, an underestimation would affect the conclusion by incorporating erroneous initial model outputs.

On the other hand, researchers often consider various spin-up time even for the same model. Lin et al (2016) considered a spin-up period of 19 days for the Xianjiang (XAJ) model during a four-month streamflow simulation for the Shiguanhe River Basin, China. In another study, Lu et al. (2008) considered only 12 h of spin-up time while forecasting floods at the Huaihe River Basin's Wangjiaba sub-basin. Even if the spin-up times are dissimilar for conceptual and physical models, this study serves important information for the XAJ model as well as other modeling communities. Firstly, it provides a basis for estimating the spin-up time for the XAJ model using widely available data sets. Secondly, it establishes a conceptual basis and shows the variations of spin-up time based on the simulation start time that provides new insights even for the physical models (the controls might be different). Thirdly, it ascertains new approaches to explore and define model spin-up time based on

broadly acceptable Mahalanobis Distance that over comes the limitations of available spin-up detection techniques.

3) Following from (1) and (2), there are systems that are known to require longer spinup times, such as deep groundwater aquifers, large surface storages, etc. Is the study model configuration capturing any of these slower processes, or purely focused on shallow soil water storage? How appropriate is this model configuration for the study basins' dominant hydrologic regime? How generally applicable are the findings if these processes are not represented? I recommend the authors provide a bit more detail on the model and its appropriateness for the study basins.

**Author response:** The studied model mainly focuses on hydrologically active soil water storage zones. This model is extensively used in humid and semi-arid regions of China and other parts of the world. The runoff formation in the XAJ model is based on the repletion of storage concept, the runoff will start to generate once the soil moisture content of the unsaturated zone reaches its field capacity, and subsequently runoff equals the rainfall excess without further loss (Zhao, 1992). The model accepts areal mean precipitation and pan evapotranspiration as the inputs and produce streamflow from the whole basin. The applicability of the XAJ model of the study basins' have been tested by Kyi, 2014. At this point this study outcome might be true for humid and semi-humid basins of USA. Commenting on the appropriateness of this conclusion at outside USA requires further verification.

4) How does the model calibration affect the results? The calibration procedure is a bit difficult to understand based on the description provided, so I recommend going into a bit more detail on what was done and how it may/may not be impacting spinup times.

**Author response:** The calibration process does not affect the result. The model calibration and exploration of model spin-up time were performed separately (please see page-5, line-11-14). The XAJ model was firstly calibrated with saturated initial condition and thereafter the daily streamflow was validated against those of the observed by taking spin-up time long enough (10 year) to avoid the effects of the initial condition. Thereafter, these calibrated parameter values were exercised for the subsequent simulations that explores the impact of initial condition.

The model calibration section has been updated with more clarification.

5) The authors present some interesting patterns in seasonality of spinup time. However, there is little discussion of the potential physical reasons for the patterns. Are these patterns simply mirroring seasonal patterns of precipitation? Are there other physical or climatic controls that might explain some of the spread, or help us determine how we can apply these results to other models or domains? I recommend expanding the discussion section to address some of these questions, which should give the paper a much broader relevance.

[Figure]

Fig. 1: Seasonality of precipitation and model spin-up time: (a) wet basins and (b) dry basins.

**Author response:** Thank you very much. You are right. Figure 1 suggests that, the seasonality of spin-up time is similar to the seasonality of precipitation. Monthly spin-up time shows strong correlation ($r=0.87$) with monthly total precipitation. The spin-up time was explored by declaring two extreme initial conditions (0% and 100% soil moisture). It is intuitive that the higher the distance between the mean condition and the initial condition, the longer the time requires reaching the equilibrium. The distance between the mean condition and any of the initial condition for the months with higher precipitation (also difference between the maximum and minimum precipitation is high) is wider as compared than that of months with lesser precipitation. Therefore, the higher the monthly precipitation, the larger the spin-up time is.

This discussion will be added into the manuscript.

Specific Comments:
Overall: The paper is nicely organized and figures are clear. However, it would benefit from additional grammar/typo editing throughout the paper.

1. Introduction: Portions of the introduction section (e.g., lines 16-27) read more like methods than introduction. I would recommend moving these specifics on the model strategy to the methods section and dedicate a bit more of the intro to addressing the study hypotheses and rationale.

**Author response:** Thank you very much. Page 3, Line 4-27 have been moved to the section 2.6: definition of model spin-up time.

Introduction section has been improved by incorporating study hypothesis and rationale.

2. Materials and Methods
2.1 Study area: Why choose only snow-free basins? I would guess because the model does not represent snowpack dynamics, but this should be stated. Why these particular 18 basins? The basins are primarily in the South-Central US (with 2 exceptions), not distributed across the US. I recommend describing the hydrologic regime in this region so we understand some of the seasonal patterns better – what is the seasonality of precipitation? Is there deep groundwater storage? What controls the runoff response?

**Author response:** The basin characteristics (including hydro-meteorological) are given in Table 1. The studied river basins were selected intentionally mainly for two reasons.

Firstly, to maintain consistency with Rahman and Lu (2015) and Rahman et al. (2015) as they discussed model-spin-up time and soil moisture memory for the same river basins. Rahman and Lu (2015) discussed model spin-up time based on a different methodologies (single year recursive simulation) that of ours (multiyear climatologies). We prefer to analyze same river basins for comparing model spin-up outcomes derived from two different methodologies. On the other hand, Rahman et al. (2015) analyzed soil moisture memory (SMM) for the same river basins. Therefore, it enables to relate the model spin-up time and soil moisture memory. Since, soil moisture autocorrelation equation (based on what SMM was estimated) does not consider snow, Rahman et al. (2015) choose snow free MOPEX basins for analysis and ultimately led the selection of studied basin for the present study.

Secondly, experience suggests that the XAJ model does not produce better results for all the MOPEX basins, particularly for the drier basins (Kyi, 2014).

2.2 Xinanjian Model: The model assumptions and configuration are fairly important for this study, so I think more detail on the model description is warranted. For example, it is not clear whether this is a lumped or distributed model when applied to the individual basins. What types of hydrologic systems does the model perform well in, and what types do poorly with this conceptual representation?

**Author response:** The XAJ model description section has been improved.

The XAJ works as a lumped model while applied to the individual basins. It works better under humid condition and poor under arid condition.

2.4 Parameters, Calibration, Validation: The calibration/validation procedure is not really described here, and is only vaguely described in the next section. I recommend expanding this section to detail the calibration/validation procedure so the reader can understand the potential sensitivity of the spinup results to the calibration.

**Author response:** Calibration and validation procedure has been clarified.

2.5 Simulation Design: Per the previous comment, it is difficult to disentangle the calibration procedure and the spinup procedure based on the description provided. I recommend separating the descriptions and clarifying the procedures.

**Author response:** Descriptions has been separated and clarified accordingly.

3. Results and Discussion: Per the general comment, I recommend adding discussion on the possible physical reasons for some of the observed patterns. As written, this section is really just results. SMM and its calculation should be better defined. The results really need to be related back to the model assumptions, climate regime, or physical basin characteristics to be relevant to other studies.

**Author response:** Result and discussion section has been improved.

References:
Cosgrove, B.A.; Lohmann, D.; Mitchell, K.E.; Houser, P.R.; Wood, E.F.; Schaake, J.C.; Robock, A.; Sheffield, J.; Duan, Q.; Luo, L.; et al. Land surface model spin-up behaviour in the North American Land Data Assimilation System (NLDAS). J. Geophys. Res. Atmos. 2003, 108, doi:10.1029/2002JD003316.

Koster, R.D., Suarez, M.J., 2001. Soil moisture memory in climate models. J. Hydrometeorol. 2, 558–570. http://dx.doi.org/10.1175/1525- 7541(2001)002<0558:SMMICM>2.0.CO;2.

Kyi, K.H., Development of a user friendly web-based Xinanjiang model with calibration support system. Unpublished M.Sc. Thesis, Dept. of Civil and Environmental Engineering, Nagaoka University of Technology, Japan,. 2014.

Lin, C.A.; Wen, L.; Lu, G.; Wu, Z.; Zhang, J.; Yang, Y.; Zhu, Y.; Tong, L. Atmospheric-hydrological modeling of severe precipitation and floods in the Huaihe River Basin, China. J. Hydrol. 2006, 330, 249–259.

Lu, G.; Wu, Z.; Wen, L.; Lin, C.A.; Zhang, J.; Yang, Y. Real-time flood forecast and flood alert map over the Huaihe River Basin in China using a coupled hydro-meteorological modeling system. *Sci. China Ser. E Technol. Sci.* **2008**, *51*, 1049–1063.

Orth, R., R. Koster, and S. Seneviratne, Inferring soil moisture memory from streamflow observations using a simple water balance model. J. Hydrometeor. 2013, 14, 1773-1790, doi:10.1175/JHM-D-12-099.1

Rahman, M.M., Lu, M., and Kyi, K. H.: Variability of soil moisture memory for wet and dry basins, J. Hydrol., 523, 107–118, doi: 10.1016/j.jhydrol.2015.01.033, 2015.

Rahman, M.M. and Lu, M.: Model spin-up behavior for wet and dry basins: a case study using the Xinanjiang model, Water, 7, 4256–4273, doi: 10.3390/w7084256, 2015.